

# Creating HiRISE digital elevation models for Mars using the open-source Ames Stereo Pipeline

Adam J. Hepburn[1,2], Tom Holt[1], Bryn Hubbard[1], and Felix Ng[2]

[1]Geography and Earth Sciences, Aberystwyth University, Llandinam Building, Penglais Campus, Aberystwyth, SY23 3DB
[2]Department of Geography, The University of Sheffield, Sheffield, S10 2TN

**Correspondence:** A.J.Hepburn (adh28@aber.ac.uk)

**Abstract.**

The present availability of sub-decametre digital elevation models on Mars—crucial for the study of surface processes—is scarce. In contrast to the globally-available but low-resolution datasets, such models enable the study of landforms <10 km in size, which is the primary scale at which geomorphic processes have been active on Mars over the last 10-20 Ma. Stereogram-

metry is a means of producing digital elevation models from stereo pairs of images. The HiRISE camera on board the Mars Reconnaissance Orbiter has captured >3000 stereo pairs at 25 cm/pixel resolution, enabling the creation of high-resolution digital elevation models (1-2 m/pixel). Hitherto, only ∼500 of these pairs have been processed and made publicly available. Existing pipelines for the production of digital elevation models from stereo-pairs, however, are built upon commercial software, rely upon sparsely-available intermediate data, or are reliant on proprietary algorithms. In this methods paper, we present

and test the output of a new pipeline for producing digital elevation models from HiRISE stereo pairs that is built entirely upon the open source NASA Ames Stereo Pipeline photogrammetric software, making use of freely available data for cartographic rectification. This pipeline is designed for simple application by researchers interested in the use of high-resolution digital elevation models. Implemented here on a research computing cluster, this pipeline can also be used on consumer-grade UNIX computers. The four output digital elevation models produced using the pipeline presented here are globally well-registered,

with accuracy similar to those of multiple digital elevation models produced elsewhere.

## 1 Introduction

After several decades of orbital study, Mars is now well served by remotely sensed imagery. This imagery enables the detailed examination and identification of features and processes globally at ∼6 m resolution, and in many places down to 0.25 m

resolution. However, Mars is poorly served by high-resolution digital elevation models (DEMs). DEMs are critical resources for the study of surface processes. Further, they enable an understanding of 3D form and process relationships otherwise only available through in-situ study. These include investigations of recurring slope linea (Schaefer et al., 2019), glacial deposits





(Hubbard et al., 2011; Conway and Balme, 2014), aeolian ridges (Berman et al., 2018; Chojnacki et al., 2014, 2017), and fluvial deposits (Voigt and Hamilton, 2018; Tirsch et al., 2018; Adler et al., 2019; Davis et al., 2018; Hamilton et al., 2018; Stack et al., 2016; Wilson et al., 2018). DEMs also fulfil a critical mission requirement when searching for potential rover landing-sites (Stack et al., 2016; Golombek et al., 2017, 2012; Grant et al., 2018). High-resolution DEMs add crucial contextual information

to other forms of remote-sensing data. DEMs then, are vital in planetary sciences where in-situ study is limited to only a handful of lander missions. Several martian DEM datasets have been produced to fulfil this requirement, but they vary dramatically in both resolution and coverage.

The Mars Orbiter Laser Altimeter (MOLA) instrument aboard the Mars Global Surveyor (MGS) is the only dedicated altimeter to orbit Mars. It provides a globally complete model of Mars's elevation, with a vertical accuracy of ∼1.5 m (Smith

et al., 2001). MOLA has proved extremely useful for the study of the largest topographic features on Mars. These features, on the scale of the Tharsis bulge and the dichotomy boundary (1000s of km), exert an influence on the regional-global scale topography of Mars, and detail large changes in Mars's early geological history (Mège and Masson, 1996; Sharp, 1973; Frey et al., 1998). However, with an along track resolution of 300 m, and an interpolated grid-spacing of ∼460 m at the equator (Smith et al., 2001), MOLA has limited potential for the study of features <10 km in scale. During the last 2 billion years,

substantial dynamic landscape evolution has occurred at or below this scale (Day and Kocurek, 2016; Andrieu et al., 2018; Banks et al., 2018; Aye et al., 2019). These features are increasingly well-covered by 2D imagery, but there is great demand for DEMs with horizontal resolution much better than MOLA's that can be combined with imagery to resolve these landscapes.

Stereogrammetry allows information in the third dimension to be determined from two overlapping 2D images, and thus DEM generation, which should be prolific for many areas on Mars given the abundance of high-resolution imagery. By using the

parallax or difference in viewing angles between two images (Figure 1), stereogrammetry can be used for not only determining the relative topographic change in a landscape, but also the elevation in absolute terms (Fenton and Herkenhoff, 2000). In the simplest sense, stereogrammetry is the estimation of 3D coordinates from the measurement of common points between two 2D images (whose 2D coordinates may or may not be known (Beyer et al., 2018)). On Earth, this technique has been used to great effect in the Arctic circle, where a constellation of satellites has returned enough stereo-imagery for widespread DEM

coverage at 2 m/pixel resolution across multiple time windows (See ArcticDEM: Noh and Howat (2017)).

The High-Resolution Stereo Camera (HRSC) instrument aboard the Mars Express is the only existing purpose-built stereogrammetric camera to orbit Mars. HRSC includes a push-broom stereo sensor that acquires images along-track and uses the spacecraft motion to record images of the surface from different perspectives (Jaumann et al., 2007; Gwinner et al., 2016). HRSC has acquired a globally-expansive series of stereogrammetric DEMs in this manner, with a best grid-resolution of ∼75

m. DEMs can also be made using higher resolution imagery; HRSC DEMs, while valuable in their own right, also serve as a resolution bridge between the coarse MOLA DEM, and DEMs produced using the higher resolution Mars Reconnaissance Orbiter Context Camera (CTX: 6 m/pixel (Malin et al., 2007)) and High Resolution Image Science experiment (HiRISE : 25-50 cm/pixel (Mcewen et al., 2007) imagery. Herein, we focus on CTX and HiRISE datasets to produce DEMs.

The primary science-mission of the CTX and HiRISE cameras is not to create DEMs, and as such they are not arranged for

the acquisition of complete stereo pairs per orbit. Instead, stereo pairs of images are acquired over separate orbits often months



apart, as and when the spacecraft's flightpath permits. Nonetheless, >3000 stereo-pairs of HiRISE imagery and >1600 stereo CTX-pairs have been captured in this way (Tao et al., 2018). Each of these CTX and HiRISE stereo pairs can be used to create DEMs at resolutions of 24 m and 1-2 m respectively (Tao et al., 2018). However, due to the time-intensive and manual nature of the DEM processing pipeline used, only ~500 of these HiRISE pairs have been processed and released on the Planetary

5   data sciences orbital data explorer (PDS ODE: Tao et al. (2018)). No CTX DEMs have been released (Tao et al., 2018). The current scarcity of HiRISE DEMs at 1-2 m resolution limits their application to studying landforms at the decametre scale.

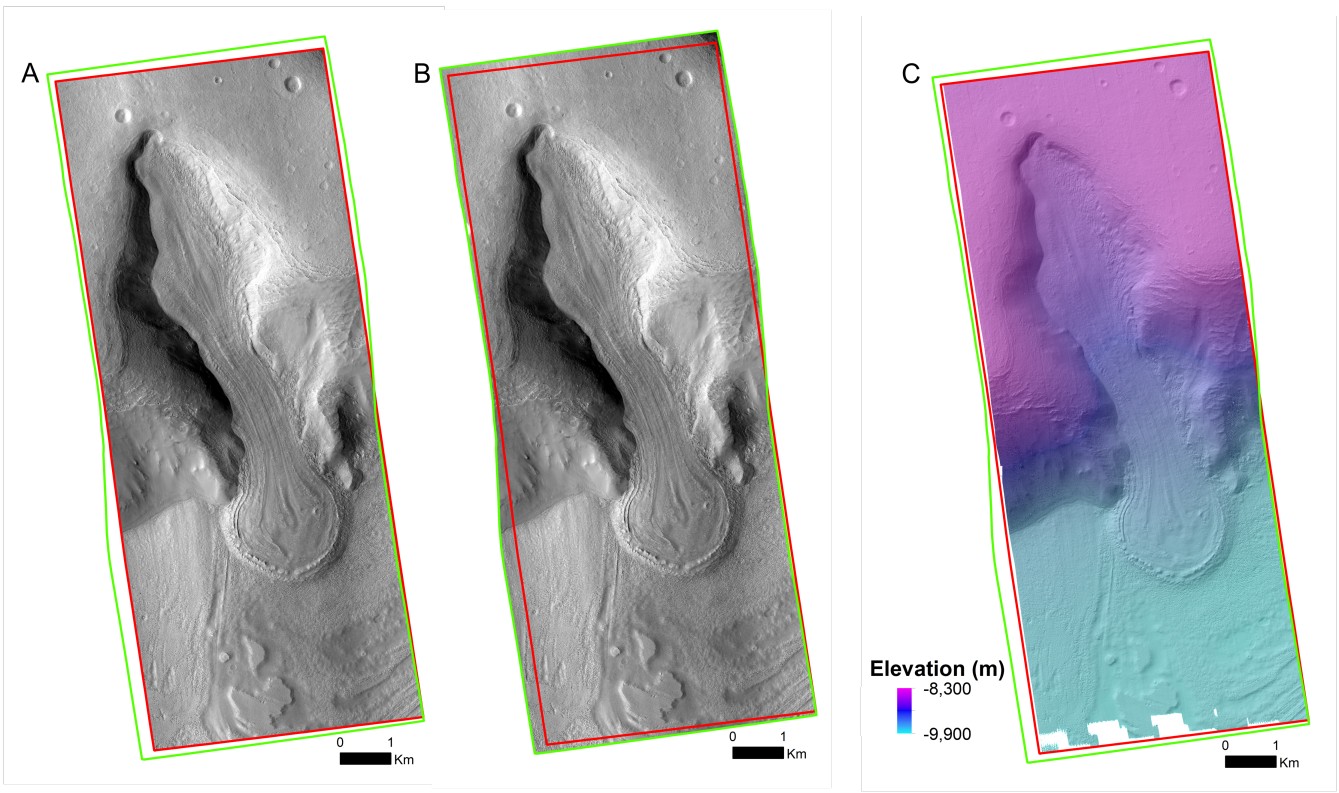

**Figure 1.** Inputs and outputs of the stereogrammetry process. Here, a left map-projected image (**A**: HiRISE EDR: ESP_052535_2225) and right map-projected image (**B**: HiRISE EDR: ESP_052324_2225) are passed to the ASP *Stereo* programme. *Stereo* uses the camera information associated with the input images to assign 3D coordinates to each cell. The output point-cloud from this process can then be converted to a DEM at 1-2 m/pixel resolution (**C**), and hillshaded to aid interpretation. The glacier-like form visible here, has been a frequent target of HiRISE imagery, but 3D models enable one to interrogate how this feature may flow downslope. This figure and all subsequent figures are orientated northwards unless it is indicated otherwise.





Given the increasing volume of stereo pairs that remain unprocessed and to better enable the utilisation of high-resolution DEMs that they can be used to create, there is a strong motivation for moving away from the traditionally esoteric means of producing DEMs towards more open-source and automatic methods (Moratto et al., 2010; Beyer et al., 2018). Accordingly, in this paper, we present and test an open-source and largely automated processing pipeline for the production of well-registered

and globally accurate DEMs from stereo pairs of HiRISE and CTX images, referred to hereafter as the Aberystwyth University pipeline (AU pipeline). Notably, our methods development work herein is focussed on the proof of concept and functionality of this new pipeline. The AU pipeline is designed to be rapidly and freely implemented by researchers, who may necessarily have past experience of DEM creation, and is based on the open-source suite of automated geodesy and stereogrammetry tools from the NASA intelligent systems division called the Ames Stereo Pipeline (ASP) (Moratto et al., 2010; Beyer et al., 2018) and the

Integrated Software for Imagers and Spectrometers version 3.5.3 programme (ISIS v3.5.3) (Anderson et al., 2004; Anderson, 2008). Several pipelines already exist for the generation of HiRISE and CTX digital elevation models, using both open source and commercial software. The AU pipeline is designed to be freely implemented, and as such, we describe only those pipelines making use of open-source software, and refer readers to Tao et al. (2018) for a summary of the existing SOCET-SET pipeline for the production of HiRISE DEMs released on the PDS ODE.

This paper is organised as follows. We first review the current state of open-source DEM production (Section 2). The ASP method for producing DEMs is described in Section 2.1, which serves as a useful overview of stereogrammetry in general, before we then go on to describe the Chicago (Mayer and Kite, 2016) and UCL CASP-GO (Tao et al., 2018) pipelines (Sections 2.2-2.3 respectively). Our new AU pipeline for the production of HiRISE DEMs is presented in Section 3. In Section 3.2 we describe how each of the DEMs produced here were evaluated, including quantitative comparison against existing DEM

products. Finally, we provided a detailed case study illustrating the production and evaluation of one DEM created using the AU pipeline (Section 4).

## 2    Existing open-source DEM generation pipelines

### 2.1    The ASP method

The first and simplest way to produce HiRISE/CTX DEMs is implemented within the ASP programme (Moratto et al., 2010;

Beyer et al., 2018). Written on the vision workbench, a general purpose image processing library built into the USGS ISIS programme (Anderson et al., 2004; Anderson, 2008), ASP allows the near-automatic processing of stereo imagery from NASA rovers and orbiters—and commercial Earth-orbiting platforms—to produce cartographic outputs (Beyer et al., 2018). As input, ASP takes the same proprietary cube (.cub) file format as ISIS. For both CTX and HiRISE images, it is necessary to convert the experimental data record (EDR) images to the .cub file format. This process is a necessary part of every pipeline discussed

herein since the reduced data record (RDR) images, which are widely available and immediately readable into GIS environments, lack the necessary geometric control for stereo processing. For CTX imagery, converting the EDR into a format suitable for stereo is a relatively simple process achieved via a sequence of ISIS commands. Image pairs are first converted to cube files using the ISIS *mroctx2isis* programme. Using *spiceinit*, the spice kernel for the cube files are then assigned, which contains





the XYZ coordinates of the camera and its orientation in 3D space. To improve the chance of stereo succeeding further down the pipeline, the images are individually radiometrically corrected using *ctxcal*. Finally, the images can be optionally destriped using *ctxevenodd*, which removes any even/odd detector striping in the image. The images can then be processed further in ASP (Beyer et al., 2018).

For HiRISE imagery this pre-processing step is more complex. Comprising 14 dual-channel charge-coupled devices (CCDs), the HiRISE EDR must be mosaicked before being passed as input to ASP (Beyer et al., 2018). The ISIS pipeline for achieving this is as follows: a) each of the 20 RED CCD channels is converted from .img to .cub using *hi2isis*, b) all 20 channels are individually radiometrically calibrated to correct for drift, instrument offset and other imperfections, using *hical*, c) *histitch* is then used to stitch together the dual channels of each CCD, forming 10 whole-CCD RED images from 20 channels, d)

spice kernels are then assigned to each CCD image using *spiceinit*, and the right ascension, declination and twist are fitted to a parabola using a least square solution with *spicefit*, as a precaution to reduce 'noisy' pointing data, e) *noproj* is used to project all 10 CCDs to the perspective of RED5, which is the central CCD, and then to modify the cube information for all other images to offset resulting distortions, f) *hijitreg* is then used to co-register adjacent CCD images based on overlapping portions, before g) all 10 CCDs are mosaicked together using *handmos* and, h) finally the resultant image is normalised using *cubenorm*.

However, the above eight-stage process is made much simpler by the ASP command *hiedr2mosaic.py*, which compiles the above ISIS routine into a single python wrapping script.

Once the HiRISE and CTX images are in the appropriate cube format, pairs are ready to be ingested directly into *Stereo*, the primary programme in the ASP suite. In practice, however, it is optimal to map project the imagery first. Map projecting the image pairs makes them as similar to one another as possible, which greatly increases the chances of subsequent pixel-matching

by the stereo algorithm (Beyer et al., 2018; Tao et al., 2018). The ASP *cam2map4stereo.py* script is based upon the ISIS *cam2map* function which takes spice kernel information and projects images according to a user-defined map projection. The ASP *cam2map4stereo.py* script does this with two input images with spice kernel information and, without further information, determines the overlap and the lowest-common resolution of each image. The two images are then map-projected at identical resolution to this shared footprint. Map projection divorces the images from their associated camera models, and their position

in 3D space is now referenced with respect to map-projected coordinates (Beyer et al., 2018). These images are then passed to *Stereo*.

At its simplest, *Stereo* takes the left and right overlapping images and by matching regions of pixels between them, outputs a point cloud that can then be further processed. *Stereo* is broken down into five steps, as follows. Stage 0 is a pre-processing step that normalises the pair with respect to each other and, for non-projected imagery, aligns the images based on one of several

available alignment algorithms. The primary purpose of this step is to remove any points present only in one image. Stage 0 outputs pre-aligned images and their respective image masks.

Stage 1 involves initial stereo correlation. The output from this stage is a disparity map, which can be thought of as a three-channel image whose pixels describe the location of a pixel in the left image and the horizontal and vertical offset between that pixel and the matching pixel in the right image. *Stereo* attempts to find a matching pixel for all values not marked invalid by





the Stage 0 masks. Correlation between large high-resolution image pairs is, therefore, computationally expensive and *Stereo* distributes the process between this and subsequent stages (Beyer et al., 2018).

Stage 1 computes an initial guess correlation using a pyramidised search that is optimised for speed but suffers on resolution. To accelerate this process, first a box filter accumulator is applied which minimises repeat correlation operations. The pyramidised approach then recursively refines the disparity estimates at increasingly fine resolutions. Each pyramidised disparity

search itself operates on sub-region tiles of comparable disparity from the preceding resolution, iteratively improving the correlation at each step (Beyer et al., 2018). The correlation window takes the form of a small rectangle that moves from the left image over the given search range in the right image. Matches between images are determined using a cost function operating in terms of disparity values, with the lowest cost match deemed best.

Following Stage 1, each pixel in the disparity map will either have estimated disparity values assigned to it, or it will have

been marked as invalid. All valid pixels are then adjusted using the sub-pixel refinement algorithm selected for Stage 2. For the quickest results, the default subpixel-mode can be used to fit a simple 2D parabola to points in an 8-pixel connected neighbourhood around each pixel in the correlation cost surface (Beyer et al., 2018). The parabola's minimum is calculated and taken as the new sub-pixel disparity value. Following this, Stage 3 addresses any outliers and matching failures. Never will every pixel be matched between a stereo pair of images. Pixels may be rejected for any combination of several reasons:

non-overlapping areas between the two images; differences in image quality, lighting, contrast, and surface specular properties; particularly smooth surfaces with a low signal-noise ratio and without a sufficient level of texture for pixel-correlation; and areas of distortion due to perspective differences between images, such as scarp walls and crater rims. Next, points with a triangulation error exceeding an automatically determined threshold are removed, and 'holes' of failed pixels are left as is.

Finally, Stage 4 converts the disparity map into a 3D point cloud. *Stereo* takes the ISIS cube information relating to the

intrinsic and extrinsic camera information from each image. Combined, these parameters enable the 'forward projection' of a 3D point onto the imaging plane of the sensor in question. Once captured, however, it is not possible to forward project the route from an image pixel to the 3D point (although it is possible to 'back-project' from the sensor to the original 3D point). These are not, however, symmetrical operations and while one camera is sufficient to "image" a 3D point on to a planar-pixel, the opposite does not hold. To back-project the route of a path from the image pixel to a 3D point, it is necessary to have two

paths from sensors both containing pixel locations which then converge at a point in 3D space. Carried out across the entire scene, these ray-intersections go on to form the basis of the output point-cloud, describing the 3D location of every matched pixel (Beyer et al., 2018).

Once the output point cloud has been generated by *Stereo*, it is then passed to the ASP *point2dem* programme. This takes the point cloud and produces a GeoTIFF 32-bit floating point DEM which can then be exported elsewhere. *Stereo* attempts to make

every pixel between a left-right pair, and the raw point clouds describe the location of each of these pixels in 3D space (Beyer et al., 2018). However, due to the nature of the *Stereo* process, and indeed more general-image matching, it is nearly impossible to match one individual pixel between two images. Instead pixel-matching matches groups of pixels within a moving window. Each pixel therefore, does contain information, but the true resolution of the output DEM depends on several factors other than input image resolution. Instead, point-clouds are generally downsampled by 3-4 times the input image resolution to improve





the reliability of pixel matching results (Beyer et al., 2018). Hence, 1-2 m/pixel DEMs can be created from 25-50 cm/pixel HiRISE images and ∼20 m/pixel DEMs from ∼ 5-6 m/pixel CTX imagery (Tao et al., 2018).

All workflows for producing DEMs from HiRISE and CTX imagery largely follow the same overall principle described above. However, to produce cartographic products fit for scientific use, it is necessary to manage the errors arising from the process. A key source of this comes from uncertainty in the camera pointing information (Tao et al., 2018). Slight errors in the

spice kernels assigned to each image mean that, in practice, rays from the two cameras do not intersect perfectly resulting in a triangulation error (Beyer et al., 2018). Generally, this error is managed by bundle-adjustment. This is the process by which the properties of the cameras and the 3D locations of the objects they image are adjusted simultaneously to minimise the error between back-projected pixel location and its actual measured location (Triggs et al., 2000). ASP, like most photogrammetric software, includes a means of bundle-adjustment (Beyer et al., 2018). If disparity exists, the ASP *bundle-adjust* tool uses a

least-squares cost-function to rectify the positioning of cameras and thousands of points across the scene (Beyer et al., 2018). Bundle adjustment helps ensure that DEMs are internally consistent and well-registered to other datasets (Beyer et al., 2018). Note that this triangulation error is not a true error, it refers only to the internal consistency of the DEM in question; high triangulation errors are always bad, whereas a low triangulation error means the DEM is internally consistent but may still be offset from the 'true elevation' of the object in question. Adjusting for this error involves incorporation of external datasets, the

geodetic locations of which have been determined *a priori* (Beyer et al., 2018).

## 2.2   The Chicago method

The 'Chicago pipeline' (Mayer and Kite, 2016) uses only ISIS and ASP routines to produce HiRISE and CTX DEMs. CTX DEMs are crucial in the Chicago pipeline as they go on to serve as an intermediate data-bridge to help coregister the HiRISE DEMs to lower-resolution datasets. CTX EDRs are prepared for ASP with ISIS routines that assign spice kernels and then

radiometrically correct the input images. The ASP bundle-adjust tool is then used to minimise triangulation error and minimise artefacts in the final DEM. These bundle-adjusted but non-map-projected images are passed to *Stereo*, which outputs a low-resolution initial DEM. The original CTX EDRs (prepared for the previous step) can then be passed to the ASP *mapproject* tool, alongside this initial low-resolution DEM onto which the images are projected. This orthorectification helps ensure *Stereo* has the maximum probability of success on steep slopes. The map-projected images are then passed to *Stereo* and *point2dem*,

producing both a full-resolution DEM and a point cloud. The final step takes the output point cloud and uses the ASP *pc_align* function to align the CTX point cloud to the MOLA point EDR (PEDR) data. A maximum displacement parameter is passed to the tool from inspection of the MOLA PEDR and the DEM from the previous step. *Pc_align* can use one of several least-cost algorithms to match two point clouds iteratively until they converge.

To produce the HiRISE DEMs, the Chicago pipeline follows the ASP pipeline described in Section 1.1, passing map-

projected images to *Stereo*. The output point cloud, however, is then passed to *pc_align*, along with the CTX-DEM created previously, taking advantage of the lower-resolution DEM to improve alignment of the HiRISE point cloud to the MOLA PEDR. As a laser-altimetry dataset, the MOLA PEDR is not subject to positioning errors of the same magnitude as other DEMs, and so is taken as the 'gold-standard' dataset to which other DEMs are aligned. The Chicago pipeline applies the





complementary approaches of bundle-adjustment and orthorectification to produce well-registered CTX DEMs, which in turn enable the production of well-registered HiRISE DEMs. However, herein lies the difficulty with the widespread implementation of this pipeline: CTX stereo-pairs are not always available for the areas covered by HiRISE pairs, and HiRISE DEMs generally have too small a footprint to intersect meaningfully with the often sparsely distributed MOLA PEDR data in the absence of a

CTX resolution-bridge.

## 2.3  The CASP-GO method

The UCL 'CASP-GO' pipeline (Tao et al., 2018) dispenses with the need for CTX DEMs entirely. This fully-automated cloud-based pipeline for the global production of CTX and HiRISE DEMs instead uses CTX orthrorectified images to coregister HiRISE pairs. To coregister these CTX pairs, orthorectified HRSC images are used. These, together with a tie-point based

multi-resolution image coregistration scheme, guarantee global geo-referencing compliance with HRSC, and by extension MOLA, DEMs. The UCL pipeline makes use of both ISIS/ASP functions—including the implementation of an initial rough disparity map and the use of the Bayes Expectation Maximisation (BEM) sub-pixel affine adaptive matcher—but then supplements and refines the outputs using *Gotcha*, a $5^{th}$ generation least squares matcher (Shin and Muller, 2012), and other proprietary algorithms. One of the stated goals of the UCL pipeline is to incorporate the CASP-GO pipeline into the online

*i-Mars* GIS project (https://bit.ly/2UrVKdT), and release processed DEMs through the PDS ODE repository. However, to date only a portion of the planned CTX DEMs have been made available on *i-Mars*, and no HiRISE DEMs have been processed. Furthermore, the underlying algorithms and their exact implementation remain unavailable to the public.

## 3  Development of near-automatic open-source AU workflow

In Section 2, we describe three open source pipelines for the production of HiRISE and CTX DEMs. However, each of these

either lacks suitable geometric control, is built upon algorithms yet to be made publicly available, or depends on data that have limited global coverage. Here, we present a new near-automatic and relatively rapid (∼72 hours) open-source AU workflow for the production of HiRISE DEMs. Making use of the globally-widespread HRSC (Jaumann et al., 2007) and MOLA DEMs (Smith et al., 2001), and existing—and open source—sub-routines, this workflow can be used to produced geometrically well-registered DEMs of a quality comparable to those released on the PDS ODE to date.

The AU pipeline we present (Figure 2) is based entirely upon the ASP 2.6.1 (available from: https://go.nasa.gov/2NIk74m) and ISIS 3.5.3 (available from: https://bit.ly/2C3jBcE) programmes. HiRISE EDRs are prepared for ingestion into ASP using the routines described above (Section 2.1), bundle adjusted, and then map-projected onto a HRSC DEM. These map-projected images are then passed to *Stereo*, and matching is carried out using the BEM subpixel refinement routine. Finally, this point-cloud is then aligned to the HRSC DEM, before being converted into the DEM format for further study. We then go on to

describe how we evaluated the output DEMs from this process, comparing them to several DEM products already available (Section 3.2).





The AU pipeline presented here is designed to enable a wide-range of researchers to create and use 1-2 m resolution HiRISE DEMs using the well-documented ASP and ISIS programmes. The AU pipeline presented here removes the need to wait for stereo-pairs to be processed and released on the PDS ODE repository. With only minor modifications, the following AU pipeline can also be used to create CTX DEMs although these are not discussed here.

### 3.1 The AU method

The first stage after pre-processing (Section 2.1) is to bundle adjust the unprojected left and right image pairs. ASP's *bundle-adjust* tool allows the input of user-selected ground control points (GCPs: Figure 3) matched between the (unprojected) left and right images, a lower-resolution, but map-projected, reference image used to assign XY coordinates to each point, and an existing DEM from which the elevation (Z) information of each GCP can be taken. For HiRISE images, a CTX EDR image and a HRSC DEM are used (for producing CTX DEMs, one would use HRSC imagery and the HRSC DEM). To ensure

coregistration between the CTX and HRSC images used in the bundle adjustment of HiRISE pairs, the CTX EDR is converted to an ASP-appropriate format and then orthorectified and map-projected onto the HRSC DEM with the ASP *map_project* tool.

GCPs are then picked on HiRISE stereo pairs. At least 30 well-distributed GCPs corresponding to identical features (such as rock outcrops and crater rims) are matched between all three images (Figure 3) using the graphical user interface version of *Stereo* (*stereo_gui*). Once GCPs are identified, the HRSC DEM is then entered, and a text file containing the 3D coordinates of

each GCP is written. This GCP list is then passed to *bundle_adjust* alongside the unprojected stereo pair.

*Bundle-adjust* is built upon the Ceres Solver—an open-source least-squares minimisation algorithm from Google—which iterates through a lowest-cost function to minimise the triangulation error between the positions of GCPs as they appear in the unprojected images, and the position of the GCPs as measured from the external georeferenced image and DEM. Uncorrected, these uncertainties would result in systematic errors in the global position of the output DEM and local slope, as well as local

distortions and warping. By bundle adjusting the 3D positions of the stereo camera as well as that of thousands of similar constraints, inconsistency between external geodetic control networks is kept to a minimum and the internal consistency of the output DEM itself is improved. *Bundle_adjust* outputs a series of transformation values, which are then passed on to *map_project* and *Stereo* in subsequent steps.

Following bundle adjustment, the images are projected using ASPs *map_project* tool. Making each image as similar as

possible to each other ensures the highest chance of success for subsequent processing. *Map_project* takes the left and right camera images together with the bundle adjustment solution and orthorectifies images onto a low resolution DEM. Here, we use the *dt4* HRSC DEM product, a map-projected 75 m/pixel DEM referenced to the Mars ellipsoid. Map-projection onto a DEM, rather than a datum, helps mitigate the impact of particularly steep terrain which is extremely difficult to match between stereo images. Map-projection leaves only very small differences in perspective between images, maximising the efficacy of

*Stereo*.

Pixel-matching between the two-map-projected images then follows the same *Stereo* procedure as described in Section 2.1, but with several additions to the command line input. The most notable of these is the selected sub-pixel refinement strategy. Rather than the default parabolic sub-pixel refinement algorithm, here we use the BEM weighted affine adaptive window



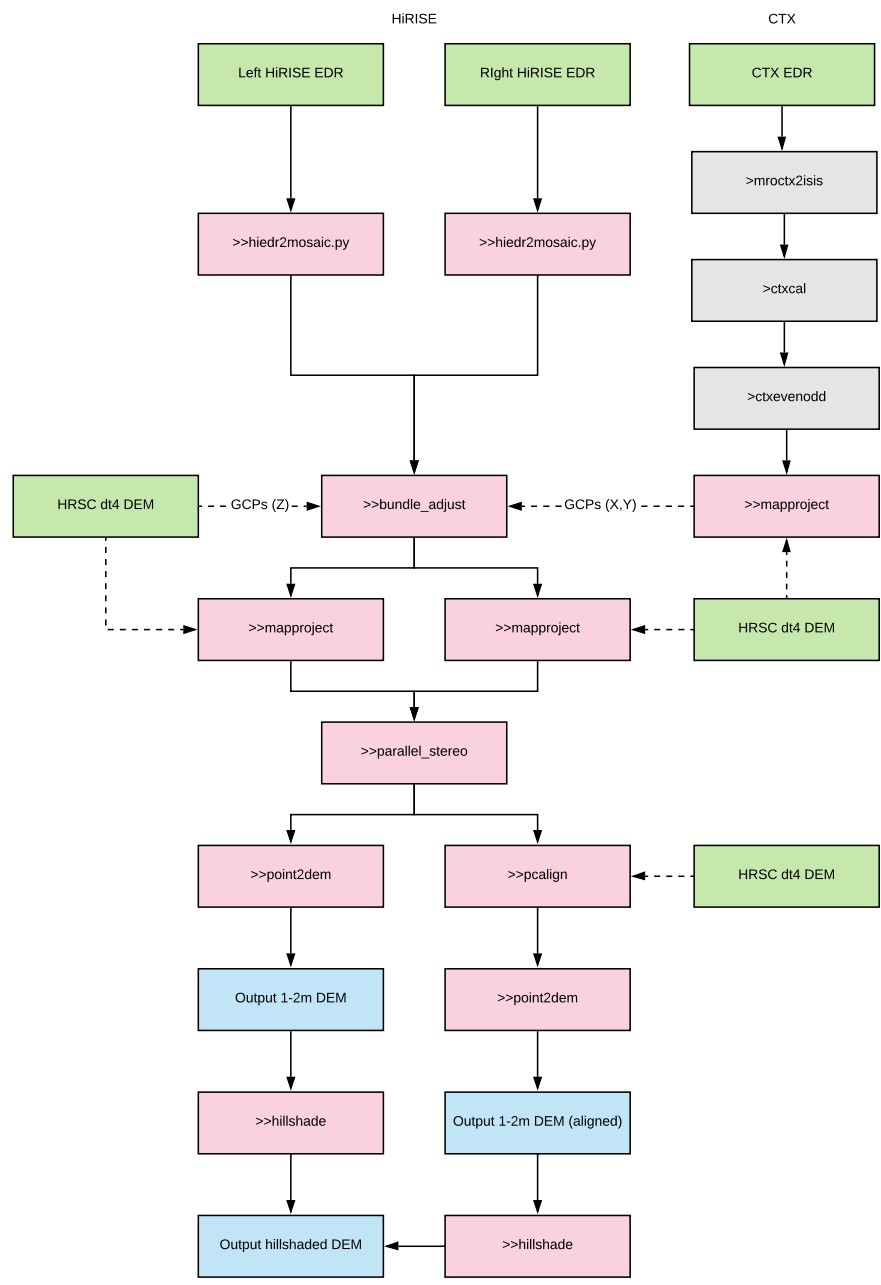

**Figure 2.** The workflow presented here for the production of 1-2 m/pixel HiRISE DEMs from stereo image pairs. Green steps represent PDS ODE data input, grey indicate ISIS commands, pink represent ASP steps, and blue represent output data.

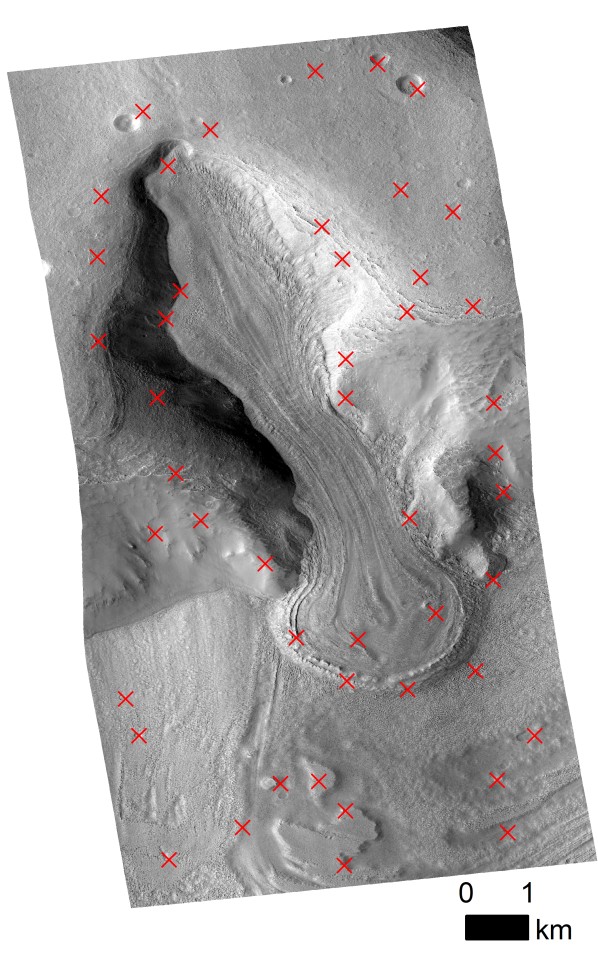

**Figure 3.** A ground-control point network (red crosses: *n=35*) selected between the left (ESP_052535_2225) and right (ESP_052324_2225) images which form AU DEM 4 (Figure 1), a map-projected CTX image (P22_009653_1487_XI_31S300W) and the HRSC DEM (h1523_0000_dt4) using *stereo_gui* in ASP. Note, points are widely distributed throughout the image.





correlation. Featuring a deformable search window which can be rotated, scaled and translated as it searches for pixels, this

adaptive algorithm is especially suited for accurate matches on steep terrain where significant foreshortening can causing perspective distortion, which is detrimental to matching. Based on the Lucas-Kanade template tracking algorithm (Baker and Matthews, 2004; Nefian et al., 2009; Beyer et al., 2018), the ASP Bayes EM treats the Lucas-Kanade variables as random parameters within an expectation maximisation (EM) framework. A Gaussian mixture component is also included, which provides a degree of robustness against image noise (Beyer et al., 2018). This Gaussian mixture component also weighs the

sub-pixel correlation kernel thereby reducing its effective size. Here, instead of the default 35 by 35 subpixel kernel window, we use a subpixel kernel size of 55-75 by 55-75 pixels to account for this weighting.

Following *Stereo*, the point cloud is converted to a DEM, and is compared to the HRSC DEM. The maximum offset between the two is then fed to *pc_align*, as an initial estimation of the magnitude of transformation necessary to align the output point cloud and the HRSC DEM. Here, with DEMs significantly offset (>100 m), we found it optimal to use two different alignment

algorithms in sequence. First, the initial transform is carried out with the Point-to-Plane Iterative Closest Point algorithm, which solves for a combination of rotation and translation. Once the point clouds are as closely aligned as possible, the transformation obtained from the previous alignment is fed to *pc_align*, and the similarity least squares transformation is then used, solving for rotation and translation alignment as well as scale. For the similarity least squares to converge, the transformation needs to be already extremely close. For point clouds that are already close prior to alignment, the Point-to-Plane ICP transformation

alone may suffice. As we go on to describe in Section 4, the use of *pc_align* does not always improve the alignment of DEMs and can degrade the alignment between the output DEM and the HRSC DEM. However, as *pc_align* outputs a new DEM and leaves the input DEM unchanged, best practice remains to make use of the tool and assess DEM outputs visually.

## 3.2 Evaluating the AU pipeline

High-resolution DEMs on Earth are generally assessed by comparison with the coordinates of fixed points within the DEM's

areal footprint. Quantitative comparison of the deviation between the surface as represented by the DEM and the surface represented by the *in-situ* points serves as a measure of DEM accuracy. However, the 'ground-truth' is largely unknown on Mars. In the pipeline presented here, GCPs are marked between stereo pairs, a third georeferenced image, and a low-resolution DEM. Yet the 3D coordinates of GCPs are themselves subject to numerous uncertainties. Therefore, they cannot be used to assess the stereo-DEMs representation of martian ground-truth. This problem applies to all Mars DEMs. In order to evaluate

the uncertainty in the DEMs produced by our pipeline we, calculated the Root-Mean Squared Error (RMSE) between HiRISE AU DEMs and those already available from the PDS ODE, both against each other and against the lower-resolution HRSC and MOLA DEMs. Four HiRISE AU DEMs were created using our pipeline and are assessed here.

Each stereo-pair was selected based on the availability of pre-existing PDS HiRISE DEM (PDS DEM hereafter) made from a different stereo-pair whose footprints partially or totally intersected those of the stereo-pair we processed. Each AU DEM

created here was made in reference to the *D_mars* datum value of 3,396.19 km. The HRSC DEMs, however, are referenced to a value of 3,396 km and so for comparison our DEMs were adjusted by 190 m. Furthermore, DEMs can be created in reference to either the datum ellipsoid or the equipotential areoid (geoid on Earth). The HRSC DEMs, for example, come in both *da4*





(areoid) and *dt4* (ellipsoid) versions. The ASP includes a tool (*dem_geoid*) for converting these datums. Conversion allows comparison between our DEMs, PDS HiRISE DEMs (referenced to a local datum and therefore effectively, the areoid), and the *da4* HRSC DEMs. In total, for the purposes of quality assessment, we produced four AU DEMs from each stereo pair;

two DEMs referenced to the ellipsoid and areoid from each of the point clouds pre and post the *pc_align* step in the workflow described above (Section 3).

### 3.2.1   Comparison 1: HiRISE-HiRISE RMSE

RMSE was first calculated between HiRISE DEMs created here, and those already available on the PDS ODE repository, using the following equation:

$$RMSE = \sqrt{(Z_{X1} - Z_{X2})^2}$$

(1)

where *X1* is the reference PDS DEM and *X2* is the comparison AU DEM produced here. DEMs were reprojected to a common equirectangular projection using a bilinear interpolation procedure. To reduce the computational strain of working with 1 m/pixel DEMs (each containing $>1.5 \times 10^8$ individual cells) DEMs were downsampled to a 10 m/pixel resolution. Downsam-

pling by this factor also helps to reduce noise and/or artefacts present in DEMs at their native 1m/pixel resolution.

Even when downsampled to 10 m resolution, differences in the projected footprint of each HiRISE DEM has the potential to greatly affect the calculated error, especially in areas of steep terrain. Further, the PDS HiRISE DEMs are subject to the same positioning uncertainties (though not necessarily of the same magnitude) as DEMs produced elsewhere, and so it may be difficult to determine which DEM is the more accurate representation. This issue is further compounded by the large resolution

gap between the HiRISE DEMs and other DEMs. This makes matching features extremely difficult, especially in highly variable topography where no clear feature may be easily distinguished. To ensure that the primary component of RMSE is vertical differences between DEMs, we excluded pixels with slopes $>10°$. Slope was calculated using the ArcGIS (v.10.5) *slope* tool, with each DEM reprojected into a length-preserving mercator projection. Slope layers were then projected back into the equirectangular projection and cell values retrieved. RMSE was calculated between PDS DEMS and the AU DEMs

produced here before and after this slope thresholding.

### 3.2.2   Comparison 2: HiRISE-HRSC RMSE

RMSE of both PDS DEM and AU DEMs were calculated against the HRSC (*da4* and *dt4*) DEMs, representing the next highest-resolution publicly available DEM product (75-250 m). Again, all DEMs were reprojected to a common equirectangular projection. Each HiRISE DEM was then downsampled to match the resolution of the HRSC DEM. The RMSE of each HiRISE DEM against the HRSC DEM could then be calculated using Equation (1) with the low-resolution reference HRSC DEM and the comparison HiRISE DEM substituted for the *X1* and *X2* terms respectively.





### 3.2.3 Comparison 3: HiRISE-MOLA RMSE

AU DEMs were also compared to MOLA individual shot data (PEDR). PDS DEMs are made in reference to this MOLA dataset, with a reported average RMSE of 1 m between the two DEMs. For comparison, the RMSE of each of our AU DEMs and the PDS DEM was calculated against the MOLA PEDR data. Each MOLA shot has a vertical uncertainty of 1.5 m, but a horizontal uncertainty of approximately 100 m. To account for this, a buffer of 50 m radius was applied to each point. Each buffered point was then rasterised to the same 10 m resolution as each of the HiRISE DEMs. The RMSE between the MOLA

PEDR and each HiRISE DEM could then be calculated using Equation (1) and substituting the MOLA PEDR for the *X1* term.

### 3.2.4 Comparison 4: MOLA-aligned HiRISE-MOLA RMES

Because PDS DEMs are made in reference to the MOLA PEDR data, the RMSE was also calculated against a AU DEM created from a point cloud aligned to the MOLA PEDR data using *pc_align*. This was compared to the RMSE of the PDS DEM against the MOLA PEDR, and the RMSE of AU DEMs made without any MOLA input.

### 3.3 DEM artefacts

Finally, visual observation of the output DEMs is a useful means of evaluating their quality. Processing artefacts—erroneous features present in a DEM that are absent in the actual terrain—are visible in both PDS and AU DEMs (Figure 4). This is predominantly due to the nature of the HiRISE images, which comprise 10 individual bands from separate CCDs (Figure 4A-B). The longitudinal lineations visible in some PDS and AU DEMs with hillshading (e.g DTEEC_028591_1880_028657_1880_l01:

https://bit.ly/2DOFAE2) is a direct result of the stitching together of these bands. Other artefacts are present in both PDS and ASP DEMs, including various failures due to lighting conditions and low-texture terrains, several examples of which are illustrated in Figure 4. Cursory visual analysis of several DEMs indicates that ASP DEMs produced here appear less affected by artefacts than HiRISE PDS DEMs. However, in areas of especially low-terrain texture, the ASP matching algorithm was less successful than the SOCET-SET pixel matcher used to create PDS DEMs (Figures 4**A** and A2). Increasing the subpixel-kernel

size can increase the efficacy of *Stereo* (Beyer et al., 2018), and although not used here, implementation of the semi-global matching algorithm in *Stereo* can also improve pixel-matching in low-texture terrains (Beyer et al., 2018).

## 4 Case study—evaluating AU DEM 2

Our workflow was successfully used to generate numerous DEMs across the surface of Mars, including four which we evaluate against other datasets. For each of these four sites (sites 1-4 hereafter), one AU DEM was generated. RMSE was calculated

between AU DEMs 1-4, a PDS DEM, HRSC DEM, and a MOLA PEDR DEM. The RMSE between AU DEMs 1-4 and preexisting PDS, HRSC and MOLA DEMs are reported in Table 1.

Site 2 is the most geologically complex landform assemblage (Figure 5), and AU DEM 2 serves as a useful illustration of common issues and details pertaining to all four AU DEMs. The discussion herein, therefore, is primarily focussed on AU



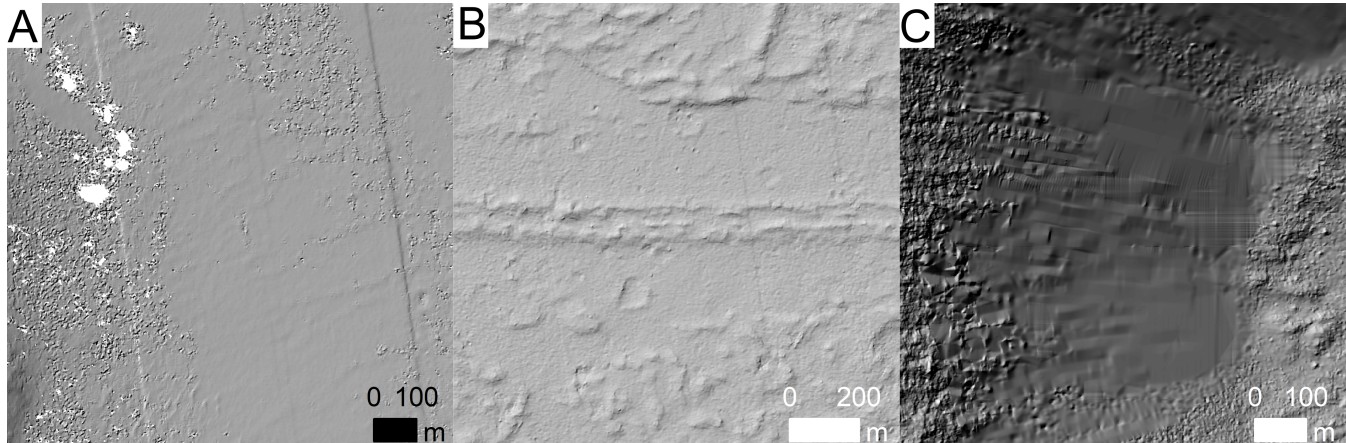

**Figure 4.** Several visible artefacts present in AU and PDS HiRISE DEMs. **A**) Pixel-matching failures associated with low-terrain texture in AU DEM 3. Longitudinal foliations are also visible. These foliations are clearly visible in either input image, but arise in the DEM because HiRISE images consist of 14-individual bands which must be stitched together to create a single image. **B**) Further longitudinal foliations associated with individual HiRISE bands, this time present in the PDS DEM for site 1 (DTEEC_002118_1510_003608_1510_A01: https://bit.ly/2tSD8Ig). **C**) Artificial smoothing present in the site 1 PDS DEM. In *lieu* of pixel-matching information, the SOCET-SET algorithm has been used to fill in the data gaps, resulting in the smooth artefact. In the ASP workflow presented here, gaps in data due to pixel-failure are left as is, leaving the empty pixels visible in **A**.

DEM 2, and for the details specific to AU DEMs 1, 3, and 4 the reader is referred to the Appendix materials. At each site, AU DEMs 1-4 compare closely to other DEMs. Further, the AU DEMs produced here represent a major increase in resolution over

5    HRSC (Figure 6A-B) and MOLA DEMs.

AU DEM 2 was created here using the HiRISE images: ESP_032115_1685 (left image) and ESP_031838_1685 (right image). This stereo pair intersects with the eastern portion of DTEEC_003910_1685_005400_1685_U01, a PDS DEM on the PDS ODE repository (https://bit.ly/2tQr89U). Both DEMs show a region of layered terrain in a Noctis Labyrinthus pit (Figure 5) with over 300 m of exposed light- and dark-toned materials, including exposed hydrated mineral units. This region

10   of Mars, towards the western limit of Valles Marineris, is of great interest to those studying the planet's aqueous history (e.g Weitz et al., 2011). The landform assemblage within site 2 consists of a flat bottomed pit containing low-amplitude aeolian landforms, flanked by a steep sided cliff-portion in the north and a more gradual slope in the south (Figures 5 and 6A-B). Site 2 includes a complex mixture of terrains, surface textures, and gradients (Figure 5). It is, therefore, an ideal environment in which to test our pipeline for the production of AU DEMs from HiRISE images.

15   Comparison one yielded a lowest RMSE of 25.8 m between *post-pc_align* AU DEM 2 and PDS DEM once slopes $>10°$ were removed (Table 1). With these steep slopes included, the RMSE increased to 39.77 m (Table 1). Before the application of *pc_align*, the RMSE of AU DEM 2 was 62.38 m and 38.54 m before and after slopes $>10°$ were removed (Table 1). AU DEM 2 before *pc_align* was horizontally well-registered to the PDS DEM (Figure 6A), and alignment shifted the AU DEM eastwards

**Table 1.** RMSE between the AU DEMs produced here, and the reference DEMs produced elsewhere. For each of the reference DEMs, RMSE is calculated against the AU DEMs *pre* (DEM) and *post* (PCA-DEM) the *pc_align* stage. For comparison, RMSE is also calculated between the PDS DEM (PDS) and the HRSC and MOLA datasets. Finally, the RMSE of the AU DEM aligned to the MOLA PEDR, against this dataset is also reported.

| Reference DEM | RMS Error (m) | | | | | | | | | | | | | | | |
|---|---|---|---|---|---|---|---|---|---|---|---|---|---|---|---|---|
| | Site 1 | | | | Site 2 | | | | Site 3 | | | | Site 4 | | | |
| | PDS | DEM | PCA-DEM | MOLA-PCA-DEM† | PDS | DEM | PCA-DEM | MOLA-PCA-DEM† | PDS | DEM | PCA-DEM | MOLA-PCA-DEM† | PDS | DEM | PCA-DEM | MOLA-PCA-DEM† |
| PDS DEM | – | 26.07 | 153.58 | – | – | 62.38 | 39.77 | – | – | 15.18 | 96.25 | – | – | 23.93 | 23.07 | – |
| PDS DEM (>10°-cutoff) | – | 24.01 | 55.68 | – | – | 38.54 | 25.80 | – | – | 14.16 | 98.71 | – | – | 9.75 | 23.42 | – |
| HRSC (*da4*) | 59.22 | 49.94 | 84.15 | – | 47.49 | 79.28 | 41.03 | – | 12.41 | 19.51 | 92.69 | – | 52.74 | 51.57 | 51.81 | – |
| HRSC (*dt4*) | – | 46.41 | 83.94 | – | – | 78.28 | 49.23 | – | – | 19.84 | 92.31 | – | – | 51.82 | 51.81 | – |
| MOLA Areoid | 15.35 | 13.19 | 87.24 | 25.23 | 46.61 | 64.63 | 41.23 | 35.32 | 7.13 | 17.59 | 89.30 | 143.58 | 10.82 | 29.43 | 26.36 | 26.10 |

† The PEDR aligned AU DEM is referenced and compared to the ellipsoidal MOLA PEDR.





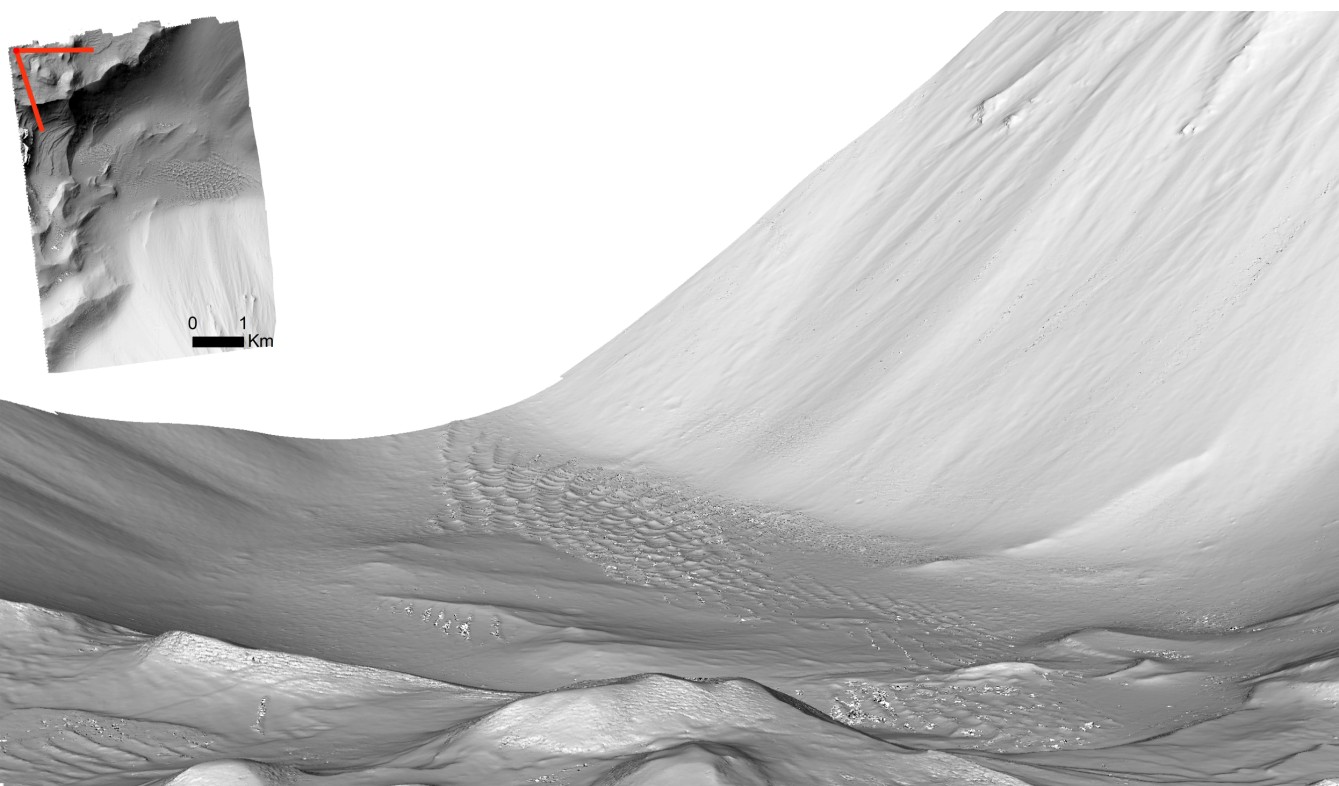

**Figure 5.** Oblique view of a region of layered terrain in a Noctis Labyrinthus pit (site 2). A complex landform assemblage is visible, including aeolian dunes, scarps, and hill-slope processes. View generated from a 2 m/pixel hillshade of AU DEM 2, draped over the AU DEM itself and visualised in *ArcScene*. Inset map shows AU DEM 2 hillshade in plan-view, approximate viewpoint marked in red.

with a strong rotational component to movement (Figure 6A E). The RMSE of both AU DEM 2 and the PDS DEM were also
5  calculated against HRSC DEM scene h1999_0000 (comparison 2). It was anticipated that the RMSE of our AU DEMs against the HRSC DEMs would be of similar magnitude to the PDS DEMs (Table 1). The *pc_align* step appears to have improved the coregristation of AU DEM 2 with the HRSC DEM, and the RMSE before and after *pc_align* was 79.28 m and 41.03 m respectively (Table 1). By comparison, the PDS DEM had an RMSE of 47.49 m against the HRSC DEM (Table 1). Finally, the RMSE of AU DEM 2 and the PDS DEM was compared against the MOLA PEDR data (Comparison 3 and 4: Table 1). As
10  before, *pc_align* reduced the RMSE from 64.63 m to 41.23 m against the reference (MOLA) dataset. The RMSE of the PDS DEM against the MOLA PEDR in reference to which it was created, was 46.61 m. RMSE fell to 35.32 m once the *pc_align* step was repeated on AU DEM 2 with the MOLA PEDR in place of the HRSC *dt4* product (Table 1).

After *pc_align*, DEM 2 had lower RMSE against the HRSC and MOLA datasets to which it was compared than the PDS DEM. *Pc_align* also lowered the RMS error between the two HiRISE products. For each of the four AU DEMs produced and tested here, an RMSE was calculated before and after point cloud alignment. As shown by Table 1, in each of the other three





**Figure 6.** Caption continued overleaf





**Figure 6.** Alignment of site 2 DEMs including the AU DEM produced using the ASP pipeline presented, the PDS DEM (DTEEC_003910_1685_005400_1685_U01: https://bit.ly/2tQr89U) and the HRSC DEM (h1999_0000_da4). **A-B**) Hillshaded PDS DEM (blue box), the AU DEM 3 *pre-pc_align* (red box), and AU DEM 3 *post-pc_align* (green box). Background is the hillshaded HRSC DEM. Note the misalignment between AU and PDS DEMs. **C**) Displacement between the AU (*pre-pc_align*) DEM and the PDS DEM. Little displacement indicates strong agreement. **D**) Displacement measured between the two AU DEMs. **E**) Displacement between the AU (*post-pc_align*) DEM and the PDS DEM. Note the strong rotational component to displacement. Panels **C-E** created using the ImGRAFT feature tracking algorithm (Messerli and Grinsted, 2015).

cases reported here, the RMSE of the ASP DEM pre-alignment was much lower than, or very similar to, the RMSE of the ASP DEM post-alignment when both are compared to the reference DEM, be it PDS HiRISE, HRSC, or MOLA data.

The RMSE of AU DEM 2 is notably higher when compared to the PDS DEM than the RMSE of the other three AU DEMs. AU DEM 2 also showed the largest decrease in RMSE when steep slopes were excluded. Together, these indicate that AU DEM 2 was poorly coregistered with the PDS HiRISE DEM to which it was compared (Figure 6E). In contrast, visual analysis of AU DEM 2 after *pc_align* indicates it is better coregistered with the HRSC DEM than the PDS DEM (Figure 6A-B), and a higher RMSE between the PDS and HRSC DEMs support this conclusion (Table 1). This demonstrates the ability of our open-source workflow to create well-registered DEMs even for challenging terrain types, in this instance working well to match pixels despite a complex landform assemblage consisting of steep, shadowed, and low-texture terrain.

AU DEM 2 was the only DEM produced here for which *pc_align* reduced the RMSE against reference datasets. Each of the other three AU DEMs all appeared well-registered without the *pc_align* step (Figures A2 and A3). In fact, in each of these cases, point cloud alignment either shifted the output DEM dramatically or not at all (Figure A4). The RMSE increase or stasis in these cases reflect this point cloud-shift or lack-thereof (Table 1). Operator discretion should be exercised when deciding whether or not to implement the final *pc_align* step in this workflow. In some instances, point cloud alignment can actually degrade the accuracy of the AU DEMs produced using our pipeline. Comparison of DEMs from multiple sources both visually and qualitatively as done here, as well as comparison to georeferenced images is, therefore, recommended.

## 5  Summary and conclusions

Our AU workflow with the open-source ASP produces DEMs of similar quality to those already available, enabling the relatively quick (∼72 hours) and largely automatic production of 1-2 m resolution DEMs in areas that at present do not have comparable high-resolution coverage. With a lowest RMSE of ∼9 m against HiRISE DEMs already publicly available, this workflow represents a valuable and easily implemented means of expanding the DEM coverage of Mars with outputs of similar quality to existing products. Further, our method is entirely based upon free open-source software and data, with no need for proprietary algorithms or intermediate CTX DEMs. This workflow has largely been implemented on the Supercomputing Wales research computing cluster, making use of the parallelised implementation of *Stereo* (*parallel_stereo*) provided in the ASP bundle. However, this workflow has been successfully run (albeit more slowly) on a consumer-grade *UNIX* laptop com-





puter with little need for parameter adjustment. By making use of open-source well-documented programmes and data the AU pipeline is relatively simple to use, and therefore ideal for the quick generation of several DEMs within a study site or area of interest.

*Code and data availability.* All code and programmes used in this study are freely available online. The latest version of the Ames Stereo Pipeline is available at: https://ti.arc.nasa.gov/tech/asr/groups/intelligent-robotics/ngt/stereo/, the Integrated Software for Imagers and Spectrometers is available at: https://isis.astrogeology.usgs.gov/, and all data used in the analysis is available from: http://ode.rsl.wustl.edu/.

Output DEMs created for this study are available from the corresponding author on request, as are example input scripts.

*Author contributions.* A.J.H performed all method work and produced the manuscript. A.J.H, T.H and B.H designed the study, and analysed the results. All authors discussed the results and commented on the writing.

*Competing interests.* The authors declare no competing interests are present

*Acknowledgements.* Thanks are extended to Samuel Doyle for his help teaching A.J.H how to use *LaTeX*. A.J.H would also like to thank

Jayne Kamintzis for **proofreading** the manuscript.



## Appendix A:  Comparison of AU DEMs 1, 3 and 4

### A1    AU DEM 1

Site 1 is the western ejecta blanket of Zumba crater, a ~3 km wide, young, and rayed impact crater in Daedalia Planum (Chuang et al., 2016). The HiRISE stereo pair ESP_020486_1510 (left) and ESP_014262_1510 (right) was used to create AU DEM 1. One PDS DEM (DTEEC_002118_1510_003608_1510_A01: https://bit.ly/2tSD8Ig) overlaps with AU DEM 1 (Figure A2). The HRSC DEM for site 1 is h2538_0000.

The RMSE of comparison 1 was 26.07 m before slopes $>10°$ were removed, and this lowered to 24.01 m once these slopes were excluded. At site 1 *pc_align* shifted the AU DEM dramatically (Figures A1 and A4). As a result, RMSE between AU 1 and the PDS DEM increased to 153.58 m, and 55.68 m once slopes $>10°$ were removed. This large increase in RMSE (and ~ 100 m difference before and after slope removal) corroborates visual analysis and indicate *pc_align* degraded the alignment of AU DEM 1 relative to the PDS DEM.

The RMSE of AU DEM 1 against the HRSC *da4* DEM was 49.94 m before and 84.15 m after *pc_align*. The RMSE of PDS DEM was 59.22 m. Together, these indicate that AU DEM 1 offered improved accuracy relative to the PDS DEM. However, it is worth noting the different footprints of each DEM, and the PDS DEM includes the full diameter of Zumba crater, AU DEM 1 only covers the western portion. A steep-walled crater represents a challenging terrain for *Stereo*, and this may account for the higher RMSE of the PDS DEM.

Finally, the RMSE against the MOLA PEDR of AU DEM 1 and the PDS DEM compare closely at 13.19 m and 15.35 m respectively. As with the other comparisons, *pc_align* increased the RMSE of the AU DEM against the MOLA PEDR. *Pc_align* did lower the RMSE of AU DEM when the MOLA PEDR was used as the alignment DEM, but at 25.23 m this was still 10 m higher than the AU DEM with no alignment.

### A2    AU DEM 3

.

Site 3 is a largely flat low-texture terrain, and includes the western rim of Endeavour Crater, a Noachian aged crater within Meridiani Planum, Arabia Terra (Mittlefehldt et al., 2018; Stein et al., 2018). AU DEM 3 was created using PSP_010341_1775 (left) and PSP_010486_1775 (right), a HiRISE image pair taken to assist in planning for the Mars Exploration Rover Opportunity's long-range traverse. DTEEC_018701_1775_018846_1775_U01 (https://bit.ly/2TrxFXX) is the site 3 PDS DEM (Figure A2), and h3198_0001 is the HRSC DEM.

At site 3, the RMSE of comparison 1 was 15.18 m, and 14.16 m once slopes $>10°$ were removed. *Pc_align* increased this RMSE to 96.25 m and 98.71 m once slopes $>10°$ were removed. The RMSE of AU DEM 3 prior to *pc_align* is the second lowest RMSE recorded against the PDS DEM reference (Table 1), suggesting the two DEMs were extremely well co-registered (Figure A2). At site 3, *pc_align* again resulted in a large shift (Figure A4) and degraded the accuracy of the AU DEM in reference to the PDS DEM.



**Figure A1.** Caption continued overleaf





**Figure A1.** Alignment of DEMs for site 1 including the AU DEM produced using the pipeline presented, the PDS DEM (DTEEC_002118_1510_003608_1510_A01: https://bit.ly/2tSD8Ig) and the HRSC DEM (h2538_0000_da4). **A-B**) Hillshaded PDS DEM (blue box), the AU DEM 3 *pre-pc_align* (red box), and AU DEM 3 *post-pc_align* (green box). Background is the hillshaded HRSC DEM. Note the large misalignment between AU and PDS DEMs after *pc_align*. **C**) Displacement between the ASP (*pre-pc_align*) DEM and the PDS DEM. Little displacement indicates strong agreement. **D**) Displacement measured between the two AU DEMs. Overlap between the two DEMs is limited to flat-featureless terrain, making pixel-matching between the two DEMs difficult. **E**) Displacement between the ASP (*post-pc_align*) DEM and the PDS DEM. As with **D**, low-texture terrain limits the efficacy of displacement tracking. Panels **C-E** created using the ImGRAFT feature tracking algorithm (Messerli and Grinsted, 2015).

AU DEM 3 also appeared well coregistered with the HRSC DEM without the *pc_align* step. Comparison 2 against the HRSC *da4* DEM yielded an RMSE of 12.41 m for the PDS DEM, 19.51 m for AU DEM 3 *pre-pc_align* and 92.69 m for AU DEM 3 *post-pc_align*. Comparisons 3 and 4 returned similar results, and the RMSE in reference to the MOLA PEDR of the PDS DEM was 7.13 m, 17.59 m for the AU DEM 3 *pre-pc_align* and 89.30 m for AU DEM 3 *post-pc_align*. *Pc_align* using the MOLA PEDR dramatically increased the RMSE (in reference to the MOLA PEDR) of the AU DEM to 143.58 m.

## A3    AU DEM 4

Site 4 is a glacier-like form in Protonilus Mensae (Figures 1, 3, and A3). A frequent HiRISE target, the glacier like-form is a striking example of the landform (Souness et al., 2012). AU DEM 4 was generated using the HiRISE image-pair ESP_052535_2225 (left) and ESP_052324_2225 (right). The site 4 PDS DEM is DTEEC_019358_2225_018857_2225_U01 (https://bit.ly/2VSdUFT), and the HRSC DEM is h1523_0000.

Site 4 is the only site at which *pc_align* did not alter the AU DEM dramatically. Compared to the PDS DEM, the RMSE of AU DEM 4 *pre-pc_align* was 23.93 m and 23.07 m after *pc_align*. Removing slopes >10° lowered the RMSE of AU DEM 4 *pre-pc_align* to 9.75 m, but for the *post-pc_align* AU DEM, RMSE increased by 0.35 m. Although the shift was subtle (Figure A4), Figure A3C indicate the AU DEM is offset by a constant 100 m in a northwest direction prior to *pc_align*. However, *pc_align* appears to have rotated AU DEM 4 clockwise (Figure A3D  E), with the centre of this rotation in the upper northwest corner. Low displacement in the upper left-hand corner (where the steepest slopes are located), removed only pixels which were in good-agreement between each DEM. In contrast, the constant offset of the *pre-pc_align* AU DEM meant that with steep slopes removed, only the low-gradient surfaces remained. Because elevation remains approximately constant across large areas where slope is low, the RMSE of the two DEMs decreased by >10 m.

Between each of the AU DEMs and the PDS DEM, the RMSE against the HRSC DEM was very similar. The RMSE of the AU DEM was 51.57 m *pre-* and 51.57 m *post-pc_align*. The RMSE of the PDS DEM was 52.74 m against the HRSC DEM. In contrast, the RMSE of the PDS DEM in reference to the MOLA PEDR was 10.82 m, whereas the RMSE of the AU DEM was 29.43 m *pre-* and 26.36 m *post-pc_align*.



**Figure A2.** Caption continued overleaf



**Figure A2.** Site 3 DEM alignment between AU DEM 3, the PDS DEM (DTEEC_018701_1775_018846_1775_U01: https://bit.ly/2TrxFXX) and the HRSC DEM (h3198_0001). **A-B**) Hillshaded PDS DEM (blue box), the AU DEM 3 *pre-pc_align* (red box), and AU DEM 3 *post-pc_align* (green box). Background is the hillshaded HRSC DEM. Note the longitudinal artefacts associated with HiRISE bands. **C**) Displacement between the AU (*pre-pc_align*) DEM and the PDS DEM. Low displacement around the crater rim indicates strong agreement. Elsewhere, the low-texture terrain hampers pixel-matching. **D**) Displacement measured between the two AU DEMs. Overlap is limited to flat-featureless terrain, making pixel-matching between the two DEMs difficult. **E**) Displacement between the ASP (*post-pc_align*) DEM and the PDS DEM. As with **D**, low-texture terrain limits the efficacy of displacement tracking. Panels **C-E** created using the ImGRAFT feature tracking algorithm (Messerli and Grinsted, 2015).





**Figure A3.** Caption continued overlead





**Figure A3.** DEM alignment at site 4 between AU DEM 4, the PDS DEM (DTEEC_019358_2225_018857_2225_U01: https://bit.ly/2VSdUFT) and the HRSC DEM (h1523_0000). **A-B**) Hillshaded PDS DEM (blue box), the AU DEM 4 *pre-pc_align* (red box), and AU DEM 4 *post-pc_align* (green box). Background is the hillshaded HRSC DEM. **C**) Displacement vector field between AU DEM 4 *pre-pc_align* and the PDS DEM. Displacement is a consistent magnitude and orientated in a northwest direction. **D**) Displacement vector field between the two AU DEMs. Displacement appears consistent, but with a lower-magnitude area in the northwestern portion of the vector-field, suggesting the misalignment was primarily rotational between the two AU DEMs. **E**) Displacement vector field between AU DEM 4 *post-pc_align* and the PDS DEM. Vector field indicates displacement with a strong rotational component. Note the low-displacement in the northwestern corner, where the steepest slopes are also located. Panels **C-E** created using the ImGRAFT feature tracking algorithm (Messerli and Grinsted, 2015).

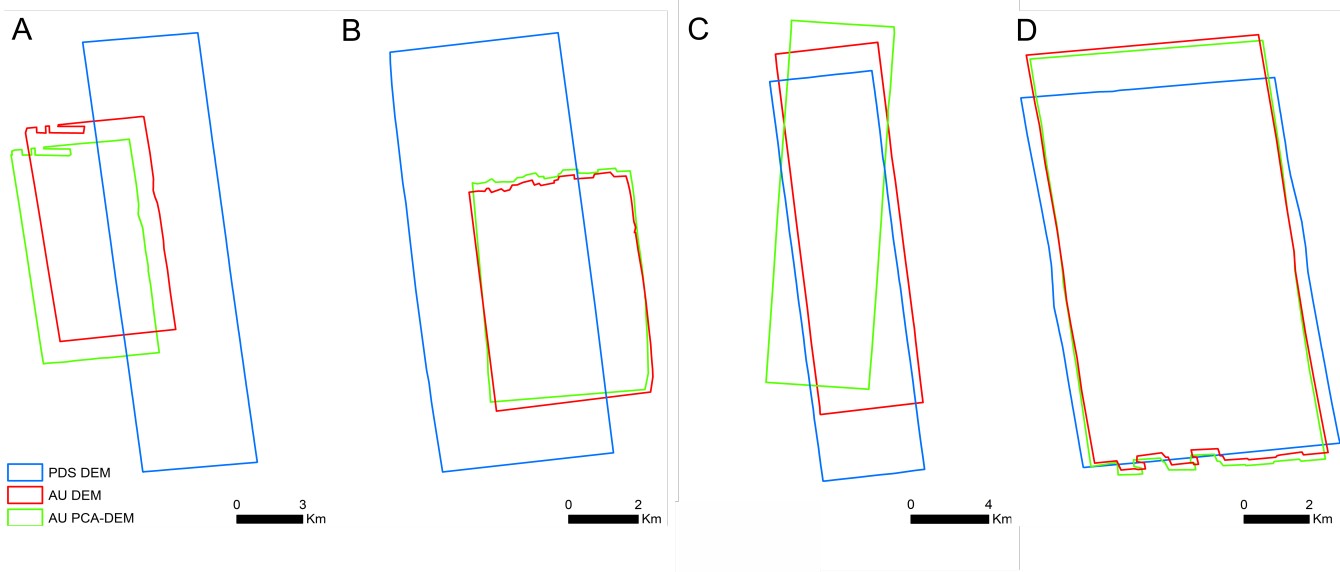

**Figure A4.** Footprints generated from each HiRISE DEM (AU and PDS) for each of the four sites. Note the large shifts after *pc_align* with the AU DEMs for sites 1 (**A**) and 3 (**B**). In each of these cases, *pc_align* dramatically increased the RMSE of AU DEMs against the reference DEMs.



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
