# Peer review of "Creating HiRISE digital elevation models for Mars using the open-source Ames Stereo Pipeline"

_Geoscientific Instrumentation, Methods and Data Systems, 2019_

## Referee Comment (RC1) · Anonymous Referee #2 · 18 Jul 2019

The manuscript of Hepburn et al. presents a modified workflow (they call it "pipeline") for the use and handling of the still published and accessible "open-source Ames Stereo Pipeline (ASP)" from the Nasa which was created to obain high resolution Digital Elevation Models (DEM's) of the Mars surface. The modifications of the authors may enable to produce faster and better evaluated high resolution DEM's. However, it is not so clearly stated, what are the benefits of the new handling concept of the still existing ASP for users and which probably are only a few specialists worldwide. Thus the authors should state more clearly who much faster and what improvements their modified pipeline will enable concerning DEM qualities.

In the Chapter 1.0 they give an introduction concerning the existing data basis (types of remote sensing images from Mars and their resolutions) as well as various methods to

calculate high resolution DEM's. This includes probably all appropriate references. Afterwards and within the Introduction chapter the authors presents an extended review (4 pages!) concerning the existing methods to calculate and produce high resolution DEMs from Mars (Chapters 1.1 to 1.3). This may be partly something to which can be introduced within a Chapter "Methodology", but not in such an extended way, since these methods are still published. Otherwise the presented manuscript should be renamed "Review of methods to create high resolution DEMs of Mars".

As a no an specialist in the details in processing new DEM's form Mars, the paper is hard to read: In the Methodology Chapter it appears often more like a user manual with many details concerning the data handling and it also contains hundreds of abbreviations, also in the figure legends where they should not be. The chapter 3 "Quality assessment: DEM comparison" sounds interesting. However, it is difficult to understand how far this is again something like a review of the authors or how far these are new results from their modified data processing. In the latter case this should appear in the Results Chapter.

The chapter 4 "Results and discussion" includes just 4 pages (2.5 pages text and a table of one page) compared to the > 5 pages chapter 1 Introduction. Both topics (results and discussion) are normally an important part of a manuscript, but here they a comparatively short. It is likely that some results of the paper are presented in the Chapters 2 and 3, but than it has to be reorganized. Furthermore, in the whole chapter 4 there is only given one reference (Beyer et al., 2018 which concerns the used NASA ASP program). This documents no or little discussion within the State of the Art. Thus this chapter again leaves the impression that the manuscript predominantly represents more a review of methods to create Martian DEM (given in Chapters 1 and 2) than important new results.

In general it is suggested that the paper requires at least major revisions. The chapters "Introduction", "Methodology" and "Results" should be reorganized. Some parts which sound like a user manual should be moved to the appendix. The review-like sections should be significantly shortened and the paper and abstract should state more clearly, what is the benefits from the modified workflow with respect to ASP handling. Otherwise the paper should become and/or state more clearly that this is a review concerning processing of Martian DEMs. Most figures and figure legend are important, but should be improved.

---

## Referee Comment (RC2) · Anonymous Referee #3 · 22 Jul 2019

In general terms, the manuscript needs a complete restructuration, in some aspects is caothic and at some areas written clearly by different technicians with different levels of knowledge without harmonisation.

My revision is mainly focused into the technical aspects due to my experience in DEMs generation, and my comments are made over the manuscript attached in this post.

On the other hand, I understand the difficulties of made a validation in a methodology when the ground truth available is of bad quality or non-existant.

Please also note the supplement to this comment:
https://www.geosci-instrum-method-data-syst-discuss.net/gi-2019-11/gi-2019-11-RC2-supplement.pdf

[Figure]

[Figure]

**Supplement:**

[revised manuscript text omitted]

---

## Author Comment (AC1) · 27 Sep 2019

Dear Editor,

We are grateful to have received two reviews for our manuscript '*Creating HiRISE digital elevation models for Mars using the open-source Ames Stereo Pipeline*'. Each reviewer provided constructive comments that have strengthened the manuscript. We respond to each of these comments below.

**Structural changes**

We were pleased that reviewer 1 is a planetary scientist, and someone we hope could make use of our pipeline. Both Reviewer 1 and Reviewer 2 raised structural concerns with the manuscript as is, and in response the paper has been reorganised to more clearly reflect that it is a methods paper. Section 1 is now where we introduce the paper, outlining the need for more accurate and well-registered digital elevation models (DEMs) of the martian surface, and how our study fits into what is already available. In Section 2 we describe three existing methods for generating open-source DEMs, providing a detailed description of stereo photogrammetry in Section 2.1. Section 3 is now where we outline and test our method. Here we also describe how our pipeline represents a useful and easily implemented means of generating well-registered DEMs using open-source software. Finally, Section 4 is where we discuss the application of our method to one case study study site, describing in detail how the output DEM compares to existing products. Throughout the main-text effort to simplify and homogenise the text has been made in response to reviewer 1's concern that the paper was hard to read in places, and reviewer 2's concern that the paper feels like disjointed writing by several authors.

**In-text changes in response to reviewer's specific comments**

**Reviewer 1**

**Comment: Who is this paper for and who will it benefit? (Reviewer 1)**

**Page 1, Lines 10-15:** The abstract has been updated to clarify the benefits of using this pipeline, and who might benefit from its use.

**Page 3, Lines 1-20**: We have also significantly changed the writing here to better elucidate the purpose of this paper, describing who might stand to benefit from the methods described therein. We have also added a paragraph at the end of Section 1 describing the organisation of the paper.

**Pages 19-20** The summary and conclusions have been changed to better reflect changes in the main text.

**Reviewer 2**

**Comment: This [Page 1, Line 19] should read 6 m spatial resolution (Reviewer 2)**

**Page 1, Line 19:** We have changed the text accordingly

**Comment: High resolution and accurate DEMs important [Page 2, Line 4] (Reviewer 2)**

**Page 1, Line 20:** We thank Reviewer 2 for this comment and have changed the text accordingly

**Comment: DEM post-spacing should be at least 5x [Page 3, Line 3]**

**Page 3, line 3**We have not changed our preferred post-spacing of 4x, noting that most studies referenced here recommend a post-spacing of 4x (e.g. Beyer et al., 2018, Tao et al., 2018). We also note that all HiRISE PDS DEMs made publicly available use a post-spacing value of 4x.

**Comment: ??? [Page 4, Line 2 ]**

**Page 3 Line 9:** We have not removed the word "esoteric" because we believe it fits well in the context of the sentence.

**Comment: A better structure of this paragraph [Page 4, Lines 15-21] would help in understanding of the contents of this paper**

**Page 3, Lines 22-27:** We have modified and simplified this paragraph to more clearly communicate the structure of the manuscript

**Comment: To be clear this paragraph must be reordered... [Page 4, Lines 24-34-Page 5, Lines 1-4]**

**Page 3, lines 30-34-Page 5, lines 1-10:** We thank Reviewer 2 for this comment and have updated the paragraph accordingly

**Comment: Why are images map-projected before they are passed to the *Stereo* programme? [Multiple times in text]**

**Page 5, Lines 20** At several points in the manuscript Reviewer 2 queries why we map-project our images and divorce them from their camera information before we pass them to *Stereo*. This step is taken both in our method and other methods because the camera information can be subject to large uncertainties, and map-projection helps to ensure the pair of images are as similar to one-another as possible, maximising the efficacy of the *Stereo* pixel-matching process. Here, we have updated the text accordingly.

**Comment: How does *point2dem* handle interpolation? [Page 6, Lines 28]**

**Page 6, Lines 28-30** We thank the reviewer for this point, and have modified the text accordingly

**Comment: This paragraph [Page 7 Lines 2-15] is out of place**

**Page 7, Lines 5-20** In response to this comment, which we agree with, we have created a new subsection (Section 2.2). This section is now where we describe the process of bundle-adjustment; a key step in the production of accurate cartographic DEMs, and not just rudimentary DEMs suited for visualisation or outreach.

**Comment: Erroneous assumption concerning Section 2.2 [Page 7, Lines 23-24 & 29-30] (Reviewer 2)**

**Section 2.3** In Section 2.3 (Formerly Section 2.2), we are describing a method from the Mayer and Kite (2016) LPSC abstract, and to aid clarity we have removed references to orthorectification, which we mistakenly referred to when we did just mean map-projection.

**Comment: Rephrasing of sentence [Page 8, Line 9]**

**Page 8, Line 14** We have updated the text, acknowledging this suggestion.

**Comment: Rephrasing and clarity suggestions [Page 8, Lines 27-28]**

**Page 8, Line 33, Page 9 Lines 1-2** We have updated the text in response to these suggestions.

**Comment: This paragraph [Page 9, Lines 5-8, & 15] is fully-correct but is in opposition to assumptions made elsewhere- related to the previous comment (Reviewer 2)**

**Page 9, Lines 10-13, & 20** This paragraph is not in contradiction to assumptions made elsewhere (see comment above), as elsewhere we are describing other workers methods for creating DEMs, and simply reporting the work of others.

**Comment: Why are DEMs you create made using a datum of 3,396.19 km, and not 3,396 km [Page 12, Lines 29-31] (Reviewer 2)**

**Page 13, Lines 1-5** We have elaborated why we chose the datum value we did, and why we changed datums to compare to the HRSC DEM. In short, we decided to use the more frequently used reference datum of 3,396.19 km for the implementation of our pipeline, and for the comparison of our products to HiRISE and MOLA DEMs. Comparison to HRSC DEM necessitated the use of the 3,396 km datum, and our output DEMs were changed accordingly. We now clarify that users are free to choose

either (and indeed any) datum value if they were to use our pipeline.

**Comment: I suggest a natural neighbour interpolation and not the bilinear scheme used here [Page 13, Lines 12-13] (Reviewer 2)**

**Page 13, Lines 15-16** We have not changed our method here, as natural neighbour interpolation is unsuitable for interpolating continuous datasets (e.g. DEMs). For the sake of completeness, we did test the natural neighbour interpolation, and found it had minimal impact ($<0.01$ m) on the DEM mean elevation ($<0.01$ m) and standard deviation, although it did introduce new artefacts.

**Comment: down sampling in the production of DEMs rather than post-production [Page 13, Line 15]**

We have not changed the text, down sampling could be carried out during the processing of DEMs, however, it was only down here to lower the computational requirements for measuring RMSE between 1 m HiRISE datasets. For most scientific use-cases, it would be beneficial to produce the highest resolution DEM possible.

**Comment: slope maps created using the DEM are also a useful means of visualising DEM artefacts [Page 14- Line 16] (Reviewer 2)**

**Page 15, Figure 4D** We have added a new panel to Figure 4, including a slope map, which is indeed a useful way of visualising artefacts in DEMs.

**Comment: Point-cloud alignment does not seem to have worked as expected and needs review [Page 16, Table 1, Page 19, Figure 6 caption](Reviewer 2)**

**Page 19, Figure 6 caption, & Lines 1-7** We are glad reviewer 2 also picked up on the unexpected performance of the point cloud alignment tool (*pc_align*) which actually degraded the alignment of DEMs in several cases. This is something we already discuss in text [Page 19, Lines 13-19] , but we have updated the Figure 6 caption in order to better reflect this. The performance of this tool, which is part of the Ames Stereo Pipeline processing suite, is beyond our control as it is an external programme. In most cases, the DEMs we produce are already well-aligned and it is not actually required.

**Comment: Where is this analysis in-text? [Page 19, Line 23] (Reviewer 2)**

**Page 16, Lines 2-7** We did not explicitly refer to the RMSE value quoted in the conclusion, which is taken straight from Table 1. Accordingly, we have updated Section 4 to ensure we make reference to this value in-text.